# Diagnostic performance of the WHO definition of probable dengue within the first 5 days of symptoms on Reunion Island

Yves Marie Diarra [1,2]*, Olivier Maillard[2], Adrien Vague[3], Bertrand Guihard[4], Patrick Gérardin[2ʘ], Antoine Bertolotti[2,5ʘ]

**1** UMR Processus Infectieux en Milieux Insulaire Tropical (CNRS 9192, INSERM U1187, IRD 249, Université de La Réunion), Sainte Clotilde, La Réunion, France, **2** INSERM, CIC 1410, Centre Hospitalier Universitaire Réunion, Saint-Pierre, La Réunion, France, **3** Service d'Accueil des Urgences, Centre Hospitalier Universitaire Réunion, Saint-Pierre, La Réunion, France, **4** Service d'Aide Médicale Urgente, Centre Hospitalier Universitaire Réunion, Saint-Denis, France, **5** Service des Maladies Infectieuses - Dermatologie, Centre Hospitalier Universitaire Réunion, Saint Pierre, La Réunion, France

ʘ These authors contributed equally to this work.
* diarrayvesmarie@gmail.com

**Data Availability Statement:** All relevant data are within the manuscript and its Supporting information files.

## Abstract

The relevance of the World Health Organization (WHO) criteria for defining probable dengue had not yet been evaluated in the context of dengue endemicity on Reunion Island. The objective of this retrospective diagnostic study was to evaluate the diagnostic performance of the 2009 WHO definition of probable dengue and to propose an improvement thereof. From the medical database, we retrieved the data of subjects admitted to the emergency department of the University Hospital of Reunion Island in 2019 with suspected dengue fever (DF) within a maximum of 5 days post symptom onset, and whose diagnosis was confirmed by a Reverse Transcriptase Polymerase Chain Reaction (RT-PCR). The intrinsic characteristics of probable dengue definitions were reported in terms of sensitivity, specificity, positive and negative likelihood ratios (LR+ and LR-), using RT-PCR as the gold standard. Of the 1,181 subjects who exhibited a positive RT-PCR, 652 (55%) were classified as probable dengue. The WHO definition of probable dengue yielded a sensitivity of 64% (95%CI 60–67%), a specificity of 57% (95%CI 52–61%), a LR + of 1.49 (95%CI 1.33–1.67), and a LR- of 0.63 (95%CI 0.56–0.72). The sensitivity and LR- for diagnosing and ruling out probable dengue could be improved by the addition of lymphopenia on admission (74% [95%CI: 71–78%] and 0.54 [95%CI: 0.46–0.63] respectively), at the cost of slight reductions of specificity and LR+ (48% [95%CI: 44–53%] and 1.42 [95%CI: 1.29–1.57], respectively). In the absence of, or when rapid diagnostic testing is unreliable, the use of the improved 2009 WHO definition of probable dengue could facilitate the identification of subjects who require further RT-PCR testing, which should encourage the development of patient management, while also optimizing the count and quarantine of cases, and guiding disease control.

**Funding:** Funding was obtained by European Regional Development Fund (RUNDENG 20201640-00222937). The funders had no role in study design, data collection and analysis, decision to publish, or preparation of the manuscript.

## Introduction

Dengue fever (DF), or dengue, is an arthropod-borne viral infection caused by the dengue virus (DENV), a member of the *Flaviviridae* family. There are four serotypes of DENV, namely DENV-1, DENV-2, DENV-3 and DENV-4. The burden of dengue is the largest among emerging or re-emerging arboviruses. According to the World Health Organization (WHO), the number of DF cases worldwide has increased drastically over the last two decades from 50,543 cases in 2000 to 2.4 million in 2010 and 5.2 million in 2019 [1]. Furthermore, regions at risk of dengue are expanding owing to global warming, and DENV transmission is no longer limited to tropical or subtropical areas; it now threatens temperate areas during hot seasons through the adaptation and resilience to climate change of Aedes mosquito vectors (*Ae. Albopictus* and *Ae. Aegypti*) [2, 3]. Indeed, the multiplication of Aedes vectors and arbovirus pathogens (including DENV) is mainly driven by temperature, which under the pattern of climate change, hastens mosquito reproduction, survival and shortens the extrinsic incubation period, while increasing virus replication [4]. In addition, the photoperiodic diapause of *Ae. Albopictus*, a mechanism of overwinter survivorship under cold and dark conditions, may increase its seasonal abundance and boost its dynamics of invasion in the Northern hemisphere [5].

In Reunion Island, a tropical French overseas department, people experienced a major epidemic in 1977, which was estimated to have affected 30% of the population, *i.e.*, around 150,000 cases [6, 7]. Then, dengue went silent for more than 25 years before re-emerging in 2004 [6]. Since then, DENV had been circulating under the pattern of sporadic cases and micro-epidemics (less than 300 cases/year) interspersed with inter-epidemic phases [8]. However, since 2018, dengue has been circulating continuously on the island. The circulating serotypes were DENV-2 in 2018, DENV-1 and DENV-2 in 2019, DENV-1, DENV-2 and DENV-3 in 2020 and DENV-1 in 2021 [9, 10]. Thus, by the end of August 2021, more than 70,000 cases had been confirmed and roughly 150,000 cases estimated over the 2018–2021 period [9, 10].

DENV infection is diagnosed through virological and the serological methods [11, 12]. The virological method is based on either viral isolation (culture), viral genome detection by RT-PCR (Reverse Transcriptase Polymerase Chain Reaction) and/or detection of AgNS1 (Nonstructural 1 Antigen). This usually applies during the viremic stage of DENV infection on a single serum collected within 1 to 5 days of symptom onset [11, 12]. The serological method discloses IgM (Immunoglobulin M) and IgG (Immunoglobulin G) antibody seroconversion. This usually applies on paired sera, one sampled during the acute phase and another sampled within the convalescence phase (15–21 days after the first serum), and one single positive serology indicates a probable case [11, 12].

To facilitate the benchmarking of data between countries and the identification of individuals potentially infected by dengue serotypes, the WHO introduced a new definition of probable dengue in 2009 [11, 12]. According to this definition, a patient was suspected of DF and could be diagnosed as probable dengue if he had lived in or traveled to a dengue-endemic area and had fever, plus at least two of the following criteria: nausea or vomiting, rash, aches and pains, positive tourniquet test, leukopenia and any warning sign (abdominal pain or tenderness, persistent vomiting, clinical fluid accumulation, mucosal bleed, lethargy or restlessness, liver enlargement >2 cm, increase in hematocrit concurrent with rapid decrease in platelet count) [11, 12]. However, the relevance of this definition (of probable dengue) proposed by the WHO had not yet been evaluated in the context of dengue endemicity on Reunion Island. The usual tools such as RT-PCR or Rapid Diagnostic Tests (RDT) require pre-screening to optimize them, on the basis of mere clinical and biological manifestations. Thus, the objective of this study was to evaluate the diagnostic performance of the 2009 WHO definition of probable dengue, using RT-PCR as the gold standard, and to propose an improvement thereof.

## Material and methods

### Study population

This retrospective diagnostic study of medical records and export of biomedical laboratory results involved the 1662 subjects admitted to the emergency department (ED) of the University Hospital of Reunion Island between 1st of January and 30th of June 2019 and meeting the inclusion criteria. Inclusion criteria were: suspected DF, maximum 5 days between reported onset of symptoms, admission and RT-PCR result. Subjects were suspected of having DF when they presented with fever and one of the following symptoms and signs: headache, retro-orbital pain, myalgia, arthralgia, rash, or pruritus. These 1662 subjects to whom the 2009 WHO definition of probable dengue was applied were taken from the 2365 subjects suspected of having DF. Depending on the inclusion period, a RDT and/or RT-PCR was performed to confirm the diagnosis in subjects suspected of having DF.

The data retrieved were sociodemographic, clinical, and biological characteristics. To improve data quality, a data entry guide was used to retrieve sociodemographic and clinical information from patient records. The biological data extracted directly from the laboratory was then merged with the sociodemographic and clinical data.

### Statistical analysis

**Bivariate analysis.** All variables of interest were cross-tabulated with the RT-PCR variable. Categorical variables were described as numbers and percentages. For these variables, the Pearson Chi-square test or Fisher's exact test was used as a test of association. Quantitative variables were described through the median, first and third quartiles. For these variables, the Wilcoxon rank sum test was used as a test of association.

**Identification of predictive factors for RT-PCR positivity not included in the WHO definition of probable dengue.** A classification tree was used to identify the variables to be included in the multivariate model aimed at predicting the outcome of the RT-PCR. Thus, variables that were statistically significantly associated with the RT-PCR variable were used to train the classification tree, after excluding the above-mentioned variables used as criteria of the 2009 WHO probable dengue definition. A classification tree with the maximum number of nodes was constructed 1,000 times to identify the tree complexity parameter, which minimizes the prediction error. Cross-validation was used to select the optimal tree by trimming the tree to a level that minimizes the prediction error. The variables of the optimal tree were introduced into a logistic regression model and further submitted to backward stepwise elimination procedure (with the possibility of adding the variables eliminated in the previous steps), based on the BIC (Bayesian Information Criterion) maximization. The continuous variables in the final model were dichotomized around standard values. With the exception of probable dengue, the other variables from the final model were added to the 2009 WHO definition to obtain an 'improved 2009 WHO probable dengue' definition. At each step, the adjustment for probable dengue variable was forced into the model.

**Diagnostic performance of probable dengue definitions.** The diagnostic performance of the 2009 WHO probable dengue definition and the new 'improved definition' were determined using RT-PCR as the gold standard. The intrinsic performance indicators were expressed as sensitivity, specificity, positive likelihood ratio (LR+) and negative likelihood ratio (LR-), along with their 95% confidence intervals (95%CI).

The threshold for statistical significance was set at 5%. The statistical analysis was performed with the R software (version 4.1.1, R core Team, Vienna, Austria, 2021).

### Ethical approval

This monocentric observational retrospective diagnostic study was conducted according to the reference methodology MR-004 from the National Commission of Informatics and Liberties (CNIL). In accordance with French regulations, this retrospective study did not require approval from an ethics committee. The EPIDENGUE database was registered in the national health data hub (n˚ F20201021104344). Non-refusal of participation was collected. Data was treated anonymously from patients' medical records.

## Results

Of the 1,662 subjects with a result for the 2009 WHO probable dengue definition, 1,181 (i.e. 529+652) had a RT-PCR result (Fig 1). Of these 1,181 subjects, 690 (58%) exhibited RT-PCR positive, and 652 (55%) were classified as probable dengue, according to the 2009 WHO definition of probable dengue (Table 1). Alternatively, of the 1,662 subjects fulfilling the improved definition, 1,181 (i.e. 412+769) had a RT-PCR result (Fig 2).

After training the classification tree on the variables associated with RT-PCR positivity (S1 Table), the optimal tree retained three variables: age, lymphocyte, and neutrophil counts.

The study population had a median age of 46.4 years and males accounted for 54% of this population (Table 1 and S1 Table). Among the subjects with positive RT-PCRs, 64% fulfilled the criteria for the definition of probable dengue (compared to 43% in RT-PCR negative subjects, p<0.001). The RT-PCR positive subjects also displayed lower median lymphocyte count and lower median neutrophil count on admission than RT-PCR negative subjects (0.60 vs 1.0 Giga/L and 3.0 versus 4.0 Giga/L, p<0.001, respectively).

As a result of the logistic regression model, individuals who were classified as probable dengue were twice as likely to be RT-PCR positive than individuals who were not classified as probable dengue when controlling the lymphocyte count. Individuals with lymphopenia were fourfold more likely to be RT-PCR positive than individuals without lymphopenia, when controlling probable dengue (Table 2).

The 2009 WHO definition of probable dengue yielded a sensitivity of 64% with a 95%CI of 60 to 67%. That means that out of 100 infected subjects, it was positive for 64. Its LR- was 0.63 with a 95%CI excluding the value 1 (Table 3). Sensitivity of the improved 2009 WHO probable dengue definition was 74% with a 95%CI of 71–78%. Its LR- was 0.54 with a 95%CI excluding the value 1 (Table 3).

## Discussion

Our improved definition of probable dengue allows a gain of 10 points in sensitivity related to the 2009 WHO probable dengue definition (74% *versus* 64%) at the cost of a slight reduction in specificity (48% *vs* 57%). In addition, its negative likelihood ratio was also smaller than that of the 2009 WHO definition (0.54 *vs* 0.63). In other words, the improved definition reduces the probability of being infected when the subject is not classified as probable dengue by this definition, compared to the 2009 WHO definition.

In the literature, few studies have evaluated the diagnostic performance of the 2009 WHO definition of probable dengue. One of the studies that did so was that of Nujum et al., in which a sensitivity of 76% (95%CI: 70–82%) was found [13]. The fact that this sensitivity differs from that found in our study may be due to the choice of the gold standard and the study population. The study by Nujum et al. used RT-PCR or IgM serology as reference tests, depending on whether the duration of the fever was less than 5 days or not [13]. However, we used only the RT-PCR as the gold standard, which restricted the window of opportunity to confirm a probable case of dengue to the first five days post symptoms onset but strived to follow the WHO

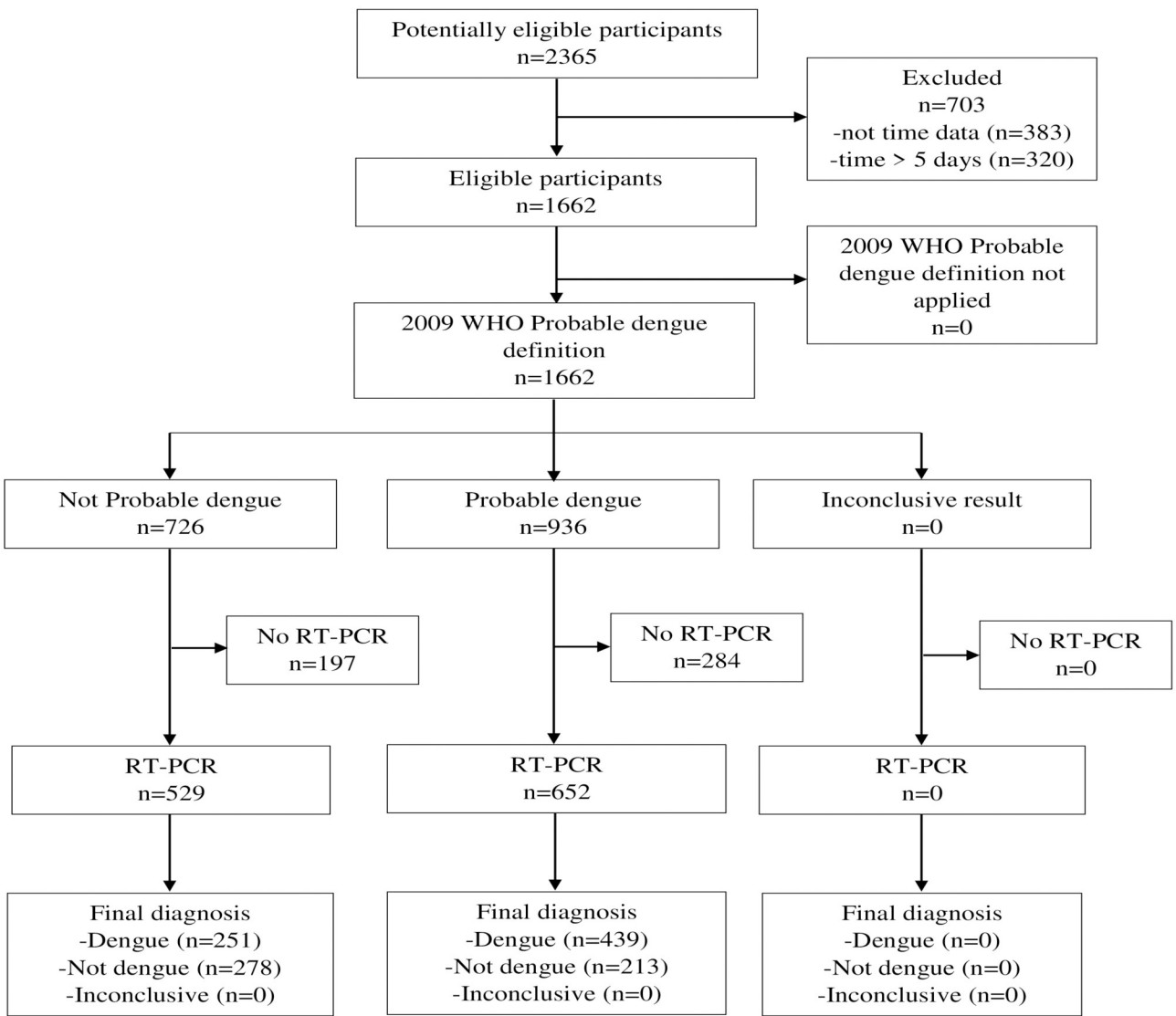

**Fig 1. STARD flow diagram 2009 WHO probable dengue definition.** WHO, World Health Organization; RT-PCR, Reverse Transcriptase Polymerase Chain Reaction.

**Table 1. Description of the study population on the variables retained by the optimal classification tree.**

| Characteristic | Overall, N = 1,181[a] | RT-PCR Negative, N = 491[a] | RT-PCR Positive, N = 690[a] | p-value[b] |
|---|---|---|---|---|
| **2009 WHO Probable dengue** | 652 (55%) | 213 (43%) | 439 (64%) | <0.001 |
| **Age (year)** | 46.43 (24.76, 69.08) | 40.32 (16.70, 68.38) | 50.03 (27.93, 70.02) | <0.001 |
| **Lymphocyte count (Giga/Litre)** | 0.70 (0.50, 1.12) | 1.00 (0.60, 1.60) | 0.60 (0.40, 0.90) | <0.001 |
| **Neutrophil count (x1,000/Microlitre)** | 3.30 (1.80, 5.50) | 4.00 (2.00, 7.10) | 3.00 (1.80, 4.68) | <0.001 |

RT-PCR, Reverse Transcriptase Polymerase Chain Reaction; WHO, World Health Organization.

[a]Median (1st quartile, 3rd quartile), count (%).

[b]Wilcoxon rank sum test, Pearson's Chi-squared test.

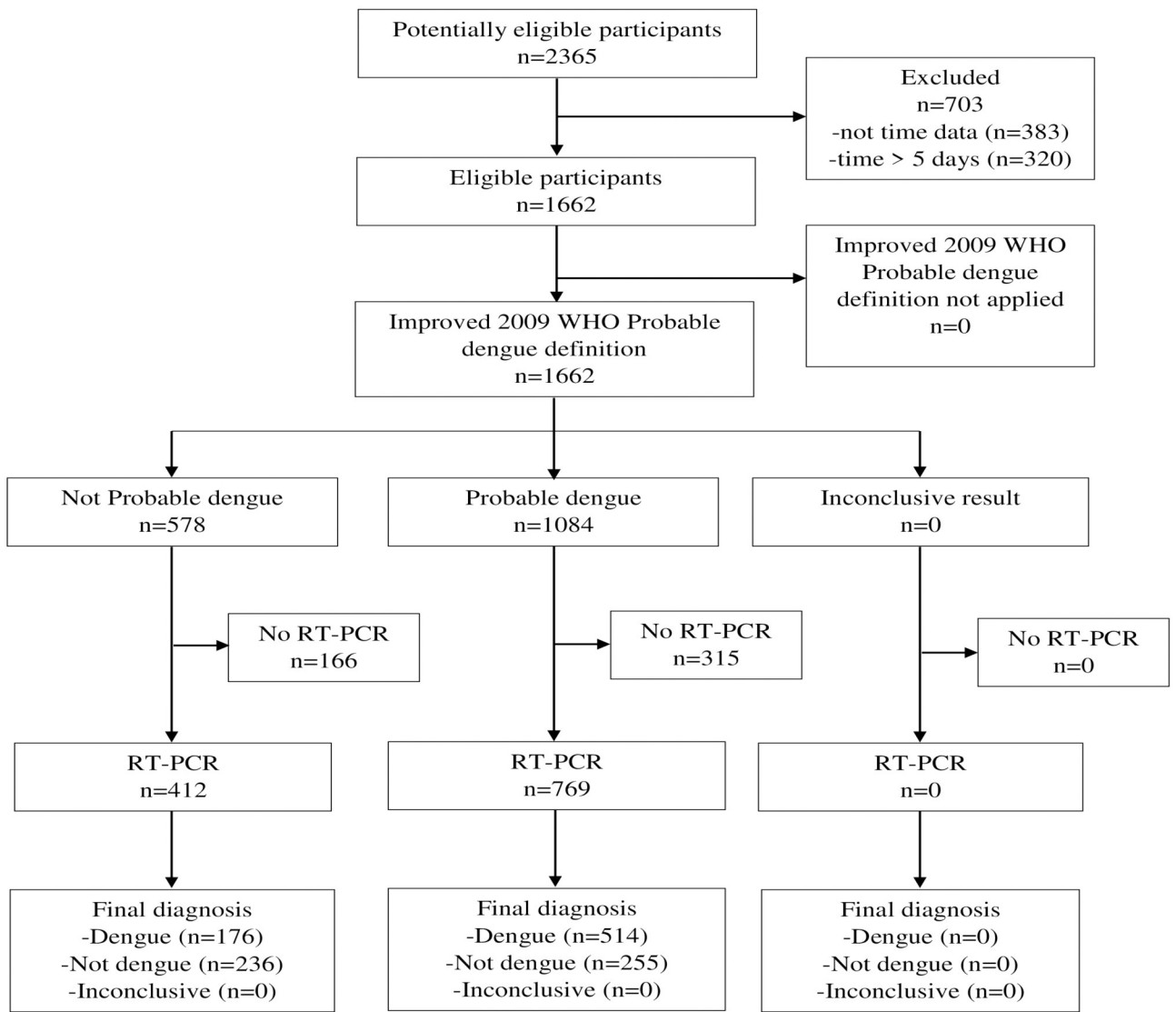

**Fig 2. STARD flow diagram Improved 2009 WHO probable dengue definition.** WHO, World Health Organization; RT-PCR, Reverse Transcriptase Polymerase Chain Reaction.

**Table 2. Independent predictive factors for RT-PCR (Reverse Transcriptase Polymerase Chain Reaction) positivity.**

| Characteristic | aOR | 95% CI | p-value |
|---|---|---|---|
| **2009 WHO Probable dengue** | 2.07 | 1.56–2.76 | <0.001 |
| **Lymphopenia (Lymphocyte count < 1.5 Giga/L)** | 4.28 | 2.92–6.34 | <0.001 |

aOR, adjusted Odds Ratio; CI, Confidence Interval; WHO, World Health Organization.

**Table 3. Diagnostic performance of probable dengue definitions.**

| Parameter | Estimate | 95% Confidence Interval |
|---|---|---|
| **2009 WHO Probable dengue** | | |
| **Sensitivity** | 64% | 60–67 |
| **Specificity** | 57% | 52–61 |
| **Positive Likelihood Ratio** | 1.49 | 1.33–1.67 |
| **Negative Likelihood Ratio** | 0.63 | 0.56–0.72 |
| **Improved 2009 WHO Probable dengue** [a] | | |
| **Sensitivity** | 74% | 71–78 |
| **Specificity** | 48% | 44–53 |
| **Positive Likelihood Ratio** | 1.42 | 1.29–1.57 |
| **Negative Likelihood Ratio** | 0.54 | 0.46–0.63 |

WHO, World Health Organization.

[a] obtained by replacing in the 2009 WHO definition of probable dengue leukopenia by 'leukopenia or lymphopenia'.

guidelines [12, 13]. This choice aimed to rule out the false positives from the IgM serology, as IgM antibodies are known to have a very significant false positive rate [14, 15]. In turn, the specificity for the 2009 WHO definition of probable dengue in the study by Nujum et al. being lower than in ours (43% *versus* 57%), confirms this [13]. Furthermore, the negative likelihood ratio found in our study was comparable to that of the Nujum et al. study (0.63 *vs* 0.60; both 95% confidence intervals overlapping) [13].

Gutiérrez et al., at the Hospital Infantil Manuel de Jesús Rivera (HIMJR), found a sensitivity of 99% (95%CI: 99–100%) for the WHO definition of probable dengue, which was significantly higher than in our study [16]. This may be due to the fact that Gutiérrez et al. combined virus isolation, RT-PCR, IgM and IgG seroconversion as the gold standard [16]. Moreover, their study is prospective whereas ours is retrospective. The presence of unspecific components in the 2009 WHO definition of probable dengue may be another reason for observing discrepancies in its performances across studies, as previously mentioned by Raafat et al. in their systematic review [17]. Thus, these authors pointed to 'elevated hematocrit along with a rapid decrease in platelet count', which could vary across studies. The differences in diagnostic performance for the 2009 WHO definition of probable dengue in our study and those in the literature may also be due to the particular context of our study, *i.e.*, well established dengue endemicity in other settings versus dengue recent endemicity on Reunion Island.

The sensitivity and negative likelihood ratio yielded by our improved 2009 WHO probable dengue definition were better than those achieved by the algorithm proposed by Caicedo-Borrero et al. (74% *vs* 65% and 0.54 *vs* 0.9, respectively) [18]. This superiority was also observed for the specificities (48 *vs* 40%). The improved definition we proposed had, however, a lower sensitivity and specificity than those found with the Daumas et al. model (74 *vs* 81%, and 48 *vs* 71%, respectively) [19]. Nevertheless, the advantage of our improved definition stands in the fact that it is based on a definition that is already in use and recommended by the WHO, and is thus more reliable and easier to implement.

The 2009 WHO definition of probable dengue detected 64 out of 100 infected individuals. The remaining 36 would not be identified, which might have a negative impact on patient management and control of the epidemic. The addition of lymphopenia detected 74 out of 100 infected patients, which represents a significant improvement in patient care and makes it easier to quarantine patients so as not to infect other mosquitoes and subsequently other individuals. Furthermore, in the context of an epidemic, when household transmission is confirmed,

it is easy to presume DF on the presence of clinical signs among other household members or among people living in the close neighborhood. The addition of the biological indicator (lymphopenia) may be useful in areas where RT-PCR is not readily available, but this addition also implies that blood has been collected, and in countries where molecular testing is readily available, this could have already included the request for a RT-PCR. Regardless of this, as the technique generally provides an answer within 24 to 72 hours, all additional means aimed at guiding the general practitioner or the ED physician in his management of the dengue patient remain legitimate and should be advocated in our health system.

One of the limitations of this study was the exclusion of subjects who did not have the timing of consultation or RT-PCR result recorded. Therefore, there is a potential for a selection bias. The use of RT-PCR as a gold standard was also another possible limitation in our study. Indeed, this test does not have a 100% sensitivity (*i.e.* it can produce false negatives), which may reduce the specificities of the probable dengue definitions that we estimated [20, 21]. Another putative limitation of this study was the use of thrombocytopenia instead of the tourniquet test (this test is not, or hardly ever, performed in practice), for which we cannot gauge any direction and magnitude of bias. However, the increased sensitivity due to thrombocytopenia of the probable dengue definition was highlighted in the paper by Nujum et al., but in which there was only one subject with a positive tourniquet test [13]. Nevertheless, this does not invalidate the interest of lymphopenia; indeed, the potential for lymphopenia has also been identified for diagnosing dengue in travelers returning from endemic areas [22].

## Conclusion

This study allowed us to evaluate the diagnostic performance of the 2009 WHO probable dengue definition and to realize the need for improving it. This led to replacing 'leukopenia' with 'leukopenia or lymphopenia' at the cost of a slight reduction in specificity. The use of the improved definition of probable dengue could therefore be useful in the setting of point of care and make it easier to identify individuals who need to further undergo RT-PCR testing, without wasting RDT resources when their provision is limited. This should foster patient management in the epidemic context, while optimizing the count and quarantine of cases and guiding disease control.

## Supporting information

**S1 Table. Description of the study population on variables associated with RT-PCR (Reverse Transcriptase Polymerase Chain Reaction).**
(DOCX)

## Acknowledgments

All authors thank the emergency units of the University Hospital of Reunion and all collaborators of the EPIDENGUE project and specifically: Azizah Issop, Jeanne Belot, Mathilde Legros, Mathys Carras, Romane Crouzet, David Hirschinger, Anne-Cecilia Etoa, Mathilde Cadic, Romain Chane-Teng, Mamitiana Randriamanana, Nolwenn Sautereau, Catherine Marimoutou and Patrick Mavingui.

## Author Contributions

**Conceptualization:** Yves Marie Diarra, Olivier Maillard, Patrick Gérardin, Antoine Bertolotti.

**Data curation:** Olivier Maillard, Adrien Vague, Bertrand Guihard.

**Formal analysis:** Yves Marie Diarra.

**Methodology:** Yves Marie Diarra, Olivier Maillard, Patrick Gérardin, Antoine Bertolotti.

**Supervision:** Patrick Gérardin, Antoine Bertolotti.

**Writing – original draft:** Yves Marie Diarra, Olivier Maillard, Patrick Gérardin, Antoine Bertolotti.

**Writing – review & editing:** Yves Marie Diarra, Olivier Maillard, Adrien Vague, Bertrand Guihard, Patrick Gérardin, Antoine Bertolotti.

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
