## [Decision Letter · Decision Letter 0]

12 Sep 2023

PONE-D-23-10419Diagnostic performance of the WHO definition of probable dengue on Reunion IslandPLOS ONE

Dear Dr. Diarra,

Thank you for submitting your manuscript to PLOS ONE. After careful consideration, we feel that it has merit but does not fully meet PLOS ONE’s publication criteria as it currently stands. Therefore, we invite you to submit a revised version of the manuscript that addresses the points raised during the review process.

Both reviewers had some very helpful suggestions for improving the manuscript.  It is my hope that the revision process will proceed much faster now that we have identified reviewers.

We look forward to receiving your revised manuscript.

Kind regards,

Kelli L. Barr, Ph.D.

Academic Editor

PLOS ONE

Journal Requirements:

"Funding was obtained by European Regional Development Fund (RUNDENG 20201640-00222937)."       

"None to declare."

Reviewers' comments:

Reviewer's Responses to Questions

**Comments to the Author**

1. Is the manuscript technically sound, and do the data support the conclusions?

Reviewer #1: Yes

Reviewer #2: Yes

2. Has the statistical analysis been performed appropriately and rigorously? 

Reviewer #1: Yes

Reviewer #2: Yes

3. Have the authors made all data underlying the findings in their manuscript fully available?

Reviewer #1: Yes

Reviewer #2: Yes

4. Is the manuscript presented in an intelligible fashion and written in standard English?

Reviewer #1: Yes

Reviewer #2: Yes

5. Review Comments to the Author

Reviewer #1: This manuscript provides useful computational information that can positively impact determining probable Dengue within the population on Reunion Island based on clinical serology characteristics. Of which, the results could be utilized with regard to patient diagnosis and management of dengue within this region.

Introduction:

I would like to see a more thorough introduction by providing clinical information on Dengue as it relates to the improvements in question in this study (i.e. model addresses lymphopenia as a factor for probable Dengue – it would be nice to demonstrate the current knowledge in the intro regarding clinical and serological presentation with regards to individuals with dengue virus).

Line 62 – Consider the addition of vector information and how a changing climate you described can impact Dengue expansion in humans

Line 65-67 - Please provide references

Line 67 – Rephrase sentence for clarity, consider changing to “before re-emerging” or “before it re-emerged”

Line 70-71 – References

Line 76 – 78 – References

Line 85 – Please explain the WHO definition for probable dengue between the sentence “Probable Dengue in 2009” and “However, the relevance….” And remove it from the methods section.

Methods:

Line 102-107 – Remove the WHO 2009 definition from methods section and place in introduction.

Line 102 Please replace the WHO definition with the actual particulars of the study participants (i.e. Study population was selected from 2365 participants and consisted of 1662 subjects that were generated from patient data from the ED based on clinical criteria associated with probable Dengue as defined by WHO…..)

Results:

Line 151 – Briefly explain the initial exclusion of the remaining 2365 that were not applicable in the study.

Discussion & Conclusion:

Both Discussion and Conclusion sections are concisely written and no changes are recommended.

Please carefully review the entire manuscript and ensure that all appropriate in-text citations are present, as there appeared to be a few missing references throughout the introduction.

Reviewer #2: Diagnostic performance of the WHO definition of probable dengue on Reunion Island

General comments

This is a good study, well done

Specific comments

Title - Would it be better to add that the study included on dengue suspects presenting within 5 days of onset of fever?

Study population - It would be good to state the definition of suspect dengue fever, who form the study population and on whom the definition of probable dengue was applied.

It will also be nice to write something about the mechanisms in place for obtaining quality data from records, since it is a retrospective study. What is the protocol in the hospital ? do all suspect cases of dengue fever undergo RT-PCR?

Table 1 : Coulumn 3, Row 1 RTPCR Negative , N= 49 is what is written, please check the number , I think last digit is missing (?491)

Discussion - Some points on relevance of using case definitions for dengue surveillance can be added and how the improvement of the case definitions helps to improve the quality of surveillance.

Best wishes to the authors.

6. PLOS authors have the option to publish the peer review history of their article (what does this mean?). If published, this will include your full peer review and any attached files.

Reviewer #1: No

Reviewer #2: **Yes: **Zinia Thajudeen Nujum

---

## [Author Response · Author response to Decision Letter 0]

1 Oct 2023

For ease of reading, responses to reviewers are shown in red in the "Response to Reviewers" file.

---

## [Editor Report · Decision Letter 1]

4 Oct 2023

PONE-D-23-10419R1Diagnostic performance of the WHO definition of probable dengue within the first 5 days of symptoms on Reunion IslandPLOS ONE

Dear Dr. Diarra,

Thank you for submitting your manuscript to PLOS ONE. After careful consideration, we feel that it has merit but does not fully meet PLOS ONE’s publication criteria as it currently stands. Therefore, we invite you to submit a revised version of the manuscript that addresses the points raised during the review process.

Both reviewers have made helpful comments for improving your paper and have identified either formatting or technical errors such as missing citations.

We look forward to receiving your revised manuscript.

Kind regards,

Kelli L. Barr, Ph.D.

Academic Editor

PLOS ONE
---

## [Author Response · Author response to Decision Letter 1]

4 Oct 2023

For ease of reading, responses to reviewers are shown in red in the "Response to

Reviewers" file

---

## [Decision Letter · Decision Letter 2]

20 Nov 2023

Diagnostic performance of the WHO definition of probable dengue within the first 5 days of symptoms on Reunion Island

PONE-D-23-10419R2

Dear Dr. Diarra,

We’re pleased to inform you that your manuscript has been judged scientifically suitable for publication and will be formally accepted for publication once it meets all outstanding technical requirements.

Kind regards,

Kelli L. Barr, Ph.D.

Academic Editor

PLOS ONE

Additional Editor Comments (optional):

Reviewers' comments:

Reviewer's Responses to Questions

**Comments to the Author**

1. If the authors have adequately addressed your comments raised in a previous round of review and you feel that this manuscript is now acceptable for publication, you may indicate that here to bypass the “Comments to the Author” section, enter your conflict of interest statement in the “Confidential to Editor” section, and submit your "Accept" recommendation.

Reviewer #1: All comments have been addressed

Reviewer #2: All comments have been addressed

2. Is the manuscript technically sound, and do the data support the conclusions?

Reviewer #1: Yes

Reviewer #2: Yes

3. Has the statistical analysis been performed appropriately and rigorously? 

Reviewer #1: Yes

Reviewer #2: Yes

4. Have the authors made all data underlying the findings in their manuscript fully available?

Reviewer #1: (No Response)

Reviewer #2: Yes

5. Is the manuscript presented in an intelligible fashion and written in standard English?

Reviewer #1: Yes

Reviewer #2: Yes

6. Review Comments to the Author

Reviewer #1: Please correct typo line 70 - (sentence refers to past information) should be "had been confirmed". The rest of the document looks good.

Reviewer #2: Best wishes to the authors

thank you for addressing the comments

I don't have any other comments to make

7. PLOS authors have the option to publish the peer review history of their article (what does this mean?). If published, this will include your full peer review and any attached files.

Reviewer #1: No

Reviewer #2: **Yes: **Zinia Thajudeen Nujum

---

## [Editor Report · Acceptance letter]

24 Nov 2023

PONE-D-23-10419R2 

Diagnostic performance of the WHO definition of probable dengue within the first 5 days of symptoms on Reunion Island 

Dear Dr. Diarra:

I'm pleased to inform you that your manuscript has been deemed suitable for publication in PLOS ONE. Congratulations! Your manuscript is now with our production department. 

Kind regards, 

on behalf of

Dr. Kelli L. Barr 

Academic Editor

PLOS ONE